**Data Availability Statement:** All data used is included in the manuscript.

# Perinatal outcome of growth restricted fetuses with abnormal umbilical artery Doppler waveforms compared to growth restricted fetuses with normal umbilical artery Doppler waveforms at a tertiary referral hospital in urban Ethiopia

**Lemi Belay Tolu** [1]*, **Roba Ararso**[1], **Abdulfetah Abdulkadir**[1], **Garumma Tolu Feyissa**[2], **Yoseph Worku**[1]

1 Saint Paul's Hospital Millennium Medical College, Addis Ababa, Ethiopia, 2 Department of Health, Behavior, and Society, Jimma University, Jimma, Ethiopia

* lemi.belay@gmail.com

## Abstract

### Background

Intrauterine growth restriction is defined as a fetal weight below the 10th percentile for a given gestational age and can be identified using umbilical artery Doppler velocimetry which is a non-invasive technique. The objective of this study was to determine the perinatal outcome of growth-restricted fetuses with abnormal umbilical artery Doppler study compared to those with normal umbilical artery Doppler waveforms at a tertiary referral hospital in Ethiopia.

### Methods

A prospective cohort study was conducted among pregnant mothers with fetal growth restriction admitted for labour and delivery from September 2018-February 2019. The data were entered and analyzed using SPSS version 23. After conducting descriptive analysis, exploring the entire data, and checking for, statistical associations between abnormal umbilical artery Doppler and outcome variables, multiple logistic regression was conducted to control for confounders.

### Results

A total of 170 pregnant mothers complicated with growth-restricted fetuses were included in the study, among which 133 were with normal umbilical artery Doppler studies and 37 were with abnormal umbilical artery Doppler studies. Four (3%) of normal and 9(24.3%) of abnormal umbilical artery Doppler studies ended in perinatal death-value = 0.001. Twenty (15%) of normal and 24(64.9%) of abnormal umbilical artery Doppler study neonates required neonatal intensive care admission-value = 0.002. Growth restricted fetuses complicated with

**Funding:** The authors received no specific funding for this work.

**Competing interests:** No authors have competing interests.

**Abbreviations:** ANC, Antenatal Care; HMD, Hyaline membrane disease; IUGR, Intra-Uterine Growth Restriction; IVH, Intraventricular hemorrhage; MAS, Meconium aspiration syndrome; MCA, Middle Cerebral Artery; NEC, Necrotizing Enterocolitis; NICU, Neonatal Intensive Care Unit; PI, Pulsatility Index; RDS, Respiratory Distress Syndrome; S/D, Systolic diastolic ratio; SGA, Small for Gestational Age; SPHMMC, Saint Paul Hospital Millennium Medical College; SPSS, Statistical Package for Social Sciences; UA, Umbilical Artery.

abnormal Doppler were two times more likely to require neonatal intensive care unit admissions compared to growth-restricted fetuses with normal umbilical artery Doppler flow, P-value 0.002, (OR = 2.059,95%CI 1.449–2.926). Growth restricted fetuses complicated with abnormal Doppler were four times more likely to end in early neonatal death compared to growth-restricted fetuses with normal umbilical artery Doppler flow, P-value 0.001, (OR = 4.136, 95%CI 3.423–4.998). However, the study is unmatched and there is a possibility of gestational age confounding the result and should be seen with the context of preterm morbidity and mortality.

## Conclusion

The abnormal umbilical artery Doppler waveform is associated with cesarean section delivery, neonatal intensive care unit admission, respiratory distress syndrome, neonatal sepsis, neonatal hyperbilirubinemia, and early neonatal death compared to normal umbilical artery Doppler flow.

## Background

Intrauterine growth restriction (IUGR) is defined as a fetal weight below the 10th percentile for a given gestational age [1, 2]. Some fetuses are constitutionally small, and they don't have an increased risk of perinatal morbidity and mortality. [1, 2]. Growth restricted fetuses who may or may not be small for the date are at increased risk of morbidity and mortality [2–4]. Identification of growth-restricted fetuses at high risk of complications is very important for management purposes. Doppler ultrasound in IUGR fetuses is used for diagnosis(differentiation of health small for date and growth-restricted fetuses) and in-utero monitoring of the progression of the disease [5]. The commonly studied and used vessels are umbilical artery(UA) and vein (UV) followed by the middle cerebral artery(MCA) [6]. The systolic/diastolic (S/D) ratio, the resistance index (RI), and the pulsatility index (PI) are the three Doppler indices most widely used to analyze arterial blood flow resistance and diagnose IUGR [2, 5, 7, 8].

Perinatal mortality rates in growth-restricted neonates are 6 to 10 times that of those with normal growth [2]. Many studies reported that respiratory distress syndrome(RDS), Necrotizing enterocolitis (NEC), Intraventricular hemorrhage (IVH), clotting disorders, and multi-organ failure are significantly more likely to occur in growth-restricted neonates [1, 8, 9]. High perinatal mortality has been reported in association more with absent and reversed end-diastolic flow velocities in the umbilical arteries [2, 5, 9–15].

The high perinatal morbidity and mortality associated with growth-restricted fetuses mandate monitoring and evaluation using different parameters. Appropriate prenatal identification and management are very important to prevent some perinatal complications that could lead to adverse outcomes in growth-restricted fetuses. It has been reported that UA Doppler significantly reduces perinatal mortality and iatrogenic premature interventions by differentiating pathologic growth restriction from constitutionally small fetuses. A metanalysis of randomized controlled studies has shown that UA Doppler in combination with standard antepartum testing, was associated with a decrease of up to 38% in perinatal mortality [16].

The aim of the current study is, therefore, to determine the perinatal outcome of growth-restricted fetuses with abnormal umbilical artery Doppler waveforms compared to normal umbilical artery Doppler waveforms at Saint Paul's Hospital Millennium Medical College.

## Materials and methods

### Study area, period, and design

This was a hospital-based prospective cohort study. The study was conducted at Saint Paul's Hospital Millennium Medical College(SPHMMC), Addis Ababa, Ethiopia from September 2018-February 2019. SPHMMC is a tertiary teaching referral hospital under the Federal Ministry of Health (FMOH). According to the statistics office of the hospital, around 10,000 deliveries were attended in 2018 and 35% of deliveries being by cesarean section. According to hospital protocol growth restriction is suspected when birth weight percentile is below 10th percentile and or femoral length to abdominal circumference is greater than 23.5%. UA artery Doppler is done for all pregnant women suspected to have a growth-restricted fetus. Sagittal view color Doppler interrogation of the free loop of the umbilical artery is used to monitor Doppler indices and diastolic flow. At each episode 2 or 3 waveforms are seen and the worst waveform is taken to inform subsequent follow-up. Those with normal Doppler flow will have weekly follow up. Those with abnormal Doppler are classified in to early (raised Doppler indices) and late (reversed or absent Doppler flow). Patients will have 1–2 times antenatal visits if the abnormality is raised Doppler indices and admitted for strict inpatient follow up if its reversed or absent umbilical artery diastolic flow. The follow up involves antepartum surveillance (biophysical profile) and umbilical artery Doppler study to evaluate fetal wellbeing and progress in the abnormality. If Doppler flow remains persistently (more than two weeks) normal the fetus is considered as constitutionally small but the follow up will continue. The target timing of delivery is 32, 34, and 37–39 weeks for reversed, absent, and raised umbilical artery Doppler flow respectively. The Doppler study is also done on the date of delivery to inform the mode of delivery. The delivery is by cesarean section for reversed and absent Doppler flow and determined by obstetric factors for those with raised indices and normal Doppler study.

We considered the following inclusion criteria: singleton intrauterine pregnancy having Antenatal care (ANC) follow up, delivery and neonatal care at SPHMMC whose gestational age was ≥28 completed weeks by reliable last normal menstrual period (LNMP) or by early ultrasound of fewer than 24 weeks, diagnosed to have IUGR according to hospital protocol. Patients with lethal congenital anomalies, intrauterine fetal death before having Doppler studies and unknown last normal menstrual period, and no ultrasound before 24 weeks were excluded. Additionally, pregnant mothers with comorbid chronic medical disorders like diabetes, severe anemia, renal disease, cardiac disease, antiphospholipid antibody syndrome, and those with known TORCH infections were also excluded. We considered the following perinatal outcomes as outcome variables: prematurity, birth weight, APGAR score, the need for resuscitation, NICU admission, RDS, neonatal sepsis, perinatal mortality. The exposure variable of interest is abnormal umbilical artery Doppler waveform compared to normal umbilical artery Doppler waveform. We considered age, place of residence, level of education, occupation, marital status, parity, gestational age, mode of delivery, and hypertension as confounding variables.

### Operational definitions

Normal UA Doppler waveform: Normal Doppler indices (less than 95th centile) and positive end-diastolic velocities.

Abnormal UA Doppler waveform: Raised (above 95th centile) indices (S/D ratio, RI, and or PI) or absent or reversed UA Doppler flow.

Prematurity: delivery after 28 weeks but before 37 weeks of gestational age.

Non-Reassuring Fetal Heart Rate Pattern (NRFHRP): abnormal fetal heart rate is considered as a non-reassuring fetal heart rate pattern in this study.

Low 5th Apgar score: 5th minute Apgar score of $< 7$.

Neonatal Intensive Care Unit(NICU) admission: those neonates admitted to NICU.

Respiratory Distress Syndrome(RDS): also known as hyaline membrane disease (HMD), is a respiratory disorder of premature babies, in this study is a clinical diagnosis considered by the neonatal care team.

Neonatal sepsis: is a type of neonatal infection and the diagnosis of which is considered by the neonatal care team clinically or confirmed microbiologically as the presence of bacterial bloodstream infections such as meningitis, pneumonia, urinary tract infection, or gastroenteritis, in the setting of fever.

Perinatal mortality: in this study its antepartum fetal death after fetal viability (above 28 weeks) intrapartum fetal death plus the death of neonates in the first seven days (early neonatal deaths) per 1000 live Birth.

Intrauterine growth restriction: birth weight below the 10th percentile for a given gestational age.

## Sample size and sampling procedure

The sample size was calculated using info stat calc version 7, for cohort study. Pregnant mothers complicated with IUGR which had abnormal UA Doppler studies were labeled as an exposed group, and pregnant mothers complicated with IUGR which had normal UA Doppler studies were labeled as a non-exposed group. Considering perinatal mortality of 28% in the exposed group and 6% in non-exposed groups [14], using the power of 80% and confidence interval (CI) of 95%, the calculated sample size was 150, adding a 10% loss to follow up gave a total sample size of 170. The ratio of non-exposed to exposed was taken as 3.6:1. So 37 cases of the exposed group and 133 cases of the non-exposed group were collected consecutively for comparison for six months.

## Data collection procedure and instrument

A structured and pretested English questionnaire was used to assess sociodemographic characteristics, obstetric factors, umbilical artery Doppler waveforms, and neonatal outcomes. Two trained midwives, who were not involved in patient care, collected data by interviewing the mother and reviewing the maternal and neonatal charts. The phone number of mothers and their card numbers were recorded for the latter tracing of neonatal outcomes.

Data collection was started at the time the women were admitted to the labor and delivery room and were continued through the intrapartum course until delivery. The neonates who were not referred to Neonatal Intensive Care Unit (NICU) were followed until mothers discharged and those neonates which were referred to NICU were followed in the NICU. The status of all neonates was checked at the seventh neonatal day. Those admitted to NICU were checked at NICU for the outcome and all those discharged home before the 7th day was checked during follow up visits. Those who didn't appear on follow up were reminded by cell phone call. Principal investigator supervised data collection and checked for completeness, accuracy, and consistency of all questionnaires.

## Data processing and analysis

Data cleaning was performed to check for outliers, missed values, and any inconsistencies before the data were analyzed using the software. Data were entered and analyzed using SPSS version 23. A chi-square test was used to check statistical associations between abnormal UA Doppler and outcome variables and covariates. Outcome variables with P value less than 0.05 were selected, and cross-tabulation was done to determine the strength and direction of the

association between abnormal UA Doppler and each outcome variable. All covariates with P value less than 0.05 (covariates associated with exposure variable) were selected for multiple regression to determine their association with each outcome variable. Statistical significance of the association between exposure and outcome variables were determined by a 95% confidence interval and p-value set at 0.05. Adjusted Risk Ratio (RR) was used to determine the strength and direction of the association between exposure and outcome variables.

## Ethical consideration

Ethical approval was obtained from Saint Paul's Hospital Millennium Medical College ethical review committee. All the datasets used and/or analyzed during the current study are included in the manuscript.

## Results

### Maternal socio-demographic characteristics of the study participants

There was a total of 5000 births managed at SPHMMC during the study period of which 170 pregnant mothers complicated with IUGR were identified. Among 170 growth-restricted fetuses, 133 were with normal UA Doppler studies and 37 were with abnormal UA Doppler studies. From abnormal UA Doppler studies, 14 of them are AED, and or REDF while 23 of them were affected Doppler indices (raised indices).

The mean gestational age at the time of diagnosis was between 34 ± 2 weeks in normal UA Doppler studies and 32 ± 2 weeks in the abnormal UA groups. The average duration of follow up is 2 weeks in the abnormal UA Doppler group and 3 weeks in the normal umbilical artery follow up. There was no loss to follow up nor missing outcome data in both groups.

There is no statistically significant difference in socio-demographic characteristics in terms of maternal age, ethnicity, religion, level of education, occupation, marital status and place of residence (See Table 1).

### Maternal reproductive and obstetric characteristics of the study participants

About 37(27.8%) of participants with normal UA Doppler were para I compared to 10(27%) of those with abnormal UA Doppler but the difference is not statistically significant. The two groups were statistically different in terms of gestational age, mode of delivery, indications for cesarean section, and hypertension. About 26(70.3%) of abnormal UA Doppler patients gave birth by cesarean section compared to 43(32.3%) of patients in the normal UA Doppler. In the abnormal UA group 11(29.7%) of the cesarean section were done for absent and or reversed end-diastolic velocity (AEDV/REDV), while 8(21.6%) of them were done for NRFHR compared to 24(18.02%) of cesarean section for NRFHR in those with normal UA. Eleven (29.7%) of the abnormal UA group had hypertension compared to 15(11.3%) of IUGR with normal UA Doppler. There is no statistically significant difference between the two groups in terms of birth weight (Table 2).

### Comparison of perinatal outcome of neonates with normal and abnormal UA Doppler waveform

All the abnormal UA Doppler waveform groups were born alive compared to one (0.75) intrapartum (stillbirth) in those with normal UA Doppler, but the result was not statistically different. Five (3.2%) of normal and 9(24.3%) of abnormal UA Doppler studies ended in perinatal death. Apgar scores of 9(6.8%) of normal and 11(29.7% of abnormal Doppler groups were less

**Table 1. Socio-demographic characteristics of mothers complicated with IUGR at SPHMMC, Addis Ababa, Ethiopia from September 2018-February 2019 (n = 133 for normal UA Doppler group, n = 37 for abnormal UA Doppler group).**

| Variable | Category | Normal UA Doppler, N (%) | Abnormal UA Doppler, N (%) | Chi-square (p-value) |
|---|---|---|---|---|
| Maternal age | <20 | 2(1.5) | 2(5.4) | |
| | 20–25 | 64(48.1) | 9(24.3) | |
| | 26–30 | 30(22.6) | 13(35.1) | 2.802(0.241) |
| | 31–35 | 19(14.3) | 9(24.3) | |
| | >35 | 9(6.8) | 1(2.7) | |
| Ethnicity | Oromo | 56(42.1) | 11(29.7) | |
| | Amhara | 34(25.6) | 12(32.4) | 0.281(0.962) |
| | Tigre | 3(0.2) | 4(10.8) | |
| | Gurage | 32(24.1) | 8(21.6) | |
| | Others | 8(6) | 2(5.4) | |
| Religion | Orthodox | 69(51.9) | 23(62.2) | |
| | Muslim | 34(25.6) | 9(24.3) | 6.405(0.063) |
| | Protestant | 30(22.6) | 5(13.5) | |
| Level of education | Illiterate | 9(6.8) | 3(8.1) | |
| | Elementary | 49(36.8) | 15(40.5) | 2.065(0.721) |
| | High school | 59(44.4) | 13(35.1) | |
| | College/university. | 16(12) | 6(16.2) | |
| Occupation | Housewife | 92(69.2) | 26(70.3) | |
| | Government employee | 14(10.5) | 5(13.5) | 4.663(0.193) |
| | Private employee | 16(12) | 6(16.2) | |
| | Merchant | 6(4.5) | 0 | |
| | Daily laborer | 4(3) | 0 | |
| | Student. | 1(0.8) | 0 | |
| Marital status | Married. | 130(97.7) | 35(94.6) | 3.402(0.431) |
| | Single. | 2(1.5) | 0 | |
| | Divorced. | 0 | 1(2.7) | |
| | Widowed. | 1(0.8) | 1(2.7) | |
| Place of residence. | Rural | 11(8.3) | 4(10.8) | 1.281(0.762) |
| | Urban | 122(91.7) | 33(89.2) | |

than seven. About 22(16.5%) of neonates with normal UA Doppler required resuscitation compared to 25(67.6%) of abnormal UA Doppler neonates. Two (5.4%) neonates with abnormal UA Doppler studies developed meconium aspiration syndrome compared to six (4.5%) of normal UA Doppler waveforms and the difference is not statistically different, P-value of 0.431.

Twenty (15%) of normal and 24(64.9%) of abnormal UA Doppler study neonates required NICU admission. Fetuses complicated with IUGR with abnormal Doppler were two times more likely to require neonatal NICU admissions compared to IUGR fetuses with normal UA Doppler flow, P-value 0.002, (OR = 2.059,95%CI 1.449–2.926). Fetuses complicated with IUGR with abnormal Doppler were four times more likely to end in END compared to IUGR fetuses with normal UA Doppler flow, P-value 0.001, (OR = 4.136, 95%CI 3.423–4.998) (Table 3).

Mode of delivery, gestational age, and hypertension was associated with abnormal UA Doppler studies (Table 2). Multiple logistic regression was done to determine the effect of those independent variables on perinatal outcomes in addition to UA Doppler abnormality. Mode of delivery and the presence of hypertension was not associated with any of the perinatal

**Table 2.** Maternal reproductive and obstetric characteristics of the pregnant mothers complicated with IUGR with normal and abnormal Doppler studies at SPHMMC, Addis Ababa, Ethiopia from September 2018-February 2019.

| Variable | Category | Normal UA Doppler, N (%) N (%) | Abnormal UA Doppler, N (%)N (%) | Chi-square (P-value) |
|---|---|---|---|---|
| **Parity.** | I | 37(27.8) | 10(27.0) | |
| | II | 6(4.5) | 4(10.8) | 2.870(0.238) |
| | III | 6(4.5) | 0 | |
| | IV | 2(1.5) | 1(2.7) | |
| | V and above | 2(1.5) | 0 | |
| **Gestational age at delivery.** | (28–32) | 2(1.5) | 2(5.4) | |
| | (32–34) | 5(3.8) | 6(16.2) | 7.283(0.007) |
| | (34–37) | 21(15.8) | 10(27.0) | |
| | 37 and above | 105(78.9) | 19(51.4) | |
| **Mode of delivery.** | Vaginal | 90(67.7) | 11(29.7) | 14.682(0.005) |
| | Cesarean delivery | 43(32.3) | 26(70.3) | |
| **Indications for cesarean delivery.** | NRFHRP | 24(18.0) | 8(21.6) | 4.532(0.023) |
| | AEDF/REDF | 0 | 11(29.7) | |
| | Mal-presentation | 7(5.3) | 0 | |
| | Dystocia. | 14(10.5) | 5(13.5). | |
| **Hypertension.** | No | 118(88.7) | 26(70.3) | 8.237(0.044) |
| | Yes | 15(11.3) | 11(29.7) | |
| **EFW** | [1000–1500] | 5(3.8) | 9(24.3) | |
| | (1500–2000] | 22(16.5) | 15(40.5) | 8.237(0.144) |
| | (2000–2500] | 75(56.4) | 12(32.4) | |
| | >2500 | 31(23.3) | 1(2.7) | |

outcomes. Gestational age is associated with NICU admission, respiratory distress syndrome (RDS), and early neonatal death. Neonates born between 28 and 32 weeks of gestational age were two times more likely to be admitted to NICU, four times more likely to have respiratory distress syndrome, and three times more likely to end up with END (Table 4).

## Discussion

The present study was conducted to compare perinatal outcomes of IUGR with normal and abnormal UA Doppler waveforms. A total of 170 pregnant mothers having a complicated IUGR were included in the study, among which 133 were with normal UA Doppler studies and 37 were with abnormal UA Doppler studies. The two groups were statistically different in

**Table 3.** Perinatal outcome of fetuses complicated with IUGR with normal and abnormal Doppler studies at SPHMMC, Addis Ababa, Ethiopia from September 2018-February 2019 ((n = 133 for normal UA Doppler group, n = 37 for abnormal UA Doppler group).

| Variable | Normal UA Doppler | Abnormal UA Doppler | Chi-square (P-value) | RR (95% CI) |
|---|---|---|---|---|
| **Stillbirth.** | 1 (0.75%) | 0 | 2.802(0.241) | - |
| Low 5th minute APGAR score. | 11 (8.3%) | 11(29.7%) | 30.475(0.001) | 2.142(1.669–2.748) |
| **The need for resuscitation.** | 22 (16.5%) | 25 (67.6%) | 9.782(0.002) | 2.350(1.648–3.352) |
| **NICU admission.** | 20 (15%) | 24 (64.9%) | 9.631(0.002) | 2.059(1.449–2.926) |
| **Respiratory distress syndrome.** | 17 (12.8%) | 19 (51.4%) | 8.001(0.005) | 2.267(1.539–3.340) |
| **Meconium aspiration syndrome.** | 6 (4.5%) | 2 (5.4%) | 3.402(0.431) | - |
| **Neonatal sepsis.** | 12 (9.0%) | 9 (24.3%) | 17.388(0.001) | 2.598(1.972–3.424) |
| **Neonatal hyperbilirubinemia** | 2 (1.5%) | 2 (5.4%) | 22.685(0.001) | 2.161(1.660–2.813) |
| **Early neonatal death(END). Died(END)** | 6 (3%) | 9 (24.3%) | 21.657(0.001) | 4.136(3.423–4.998) |

**Table 4. Multiple logistic regression of perinatal outcomes with the mode of delivery, gestational age, and hypertension among mothers complicated with IUGR with normal and abnormal Doppler studies at SPHMMC, Addis Ababa, Ethiopia from September 2018-February 2019.**

| Perinatal outcome. | Independent variable. | P-value. | Adjusted RR (95% CI) |
|---|---|---|---|
| Low 5[th] minute Apgar score | Mode of delivery | 0.391 | 2.009(0.409–9.878) |
| | Hypertension | 0.815 | 0.865(0.090–52.439) |
| | Gestational age | 0.773 | 1.023(0.124–3.336) |
| Early neonatal death. | Mode of delivery | 0.998 | 2.583(0.007–73941) |
| | Hypertension | 0.496 | 3.407(0.100–11.590) |
| | Gestational age | 0.025 | 2.103(2,048–9.884) |
| Neonatal hyperbilirubinemia | Mode of delivery | 0.998 | 7.443(0.704–4.951) |
| | Hypertension | 0.998 | 2.2730.412–368.583) |
| | Gestational age | 0.921 | 1.000(0.219–16.886) |
| The need for resuscitation. | Mode of delivery | 0.998 | 1.000(0.257–2.917) |
| | Hypertension | 0.998 | 1.000(0.090–32.439) |
| | Gestational age | 0.921 | 1.000(0.072–63.987) |
| Neonatal sepsis. | Mode of delivery | 0.200 | 0.200(0.024–1.683) |
| | Hypertension | 0.953 | 0.953(0.140–6.483) |
| | Gestational age | 0.430 | 0.430(0.063–4.505) |
| NICU admission. | Mode of delivery | 0.142 | 4.640(0.597–36.061) |
| | Hypertension | 0.225 | 2.509(0.568–11.079) |
| | Gestational age | 0.035 | 3.425(1.219–5.886) |
| Respiratory distress syndrome | Mode of delivery | 0.921 | 3.407(0.100–11.590) |
| | Hypertension | 0.773 | 2.059(1.449–2.926) |
| | Gestational age | 0.012 | 4.136(3.423–4.998) |

terms of gestational age, mode of delivery, indications for cesarean section, and presence of hypertension. In this study, 21.1% of the preterm deliveries had normal UA Doppler studies, whereas 48.6% of the preterm deliveries had abnormal UA Doppler studies. Comparing to the previous study, the percentage of preterm deliveries with normal UA Doppler study was higher (21.1% versus 14%), but the percentage of preterm deliveries with abnormal UA Doppler studies was less (48.6% versus 96%) [9].

About 70.3% of abnormal UA Doppler patients gave birth by cesarean section compared to 32.3% of patients in the normal UA Doppler. Most (29.7%) of the cesarean section in abnormal UA group was done for absent and or reversed end-diastolic velocity (AEDV/REDV), while 21.6% of them were done for NRFHR compared to 18.02% of cesarean section for NRFHR in those with normal UA. About 29.7% of the abnormal UA group had hypertension compared to 11.3% of IUGR with normal UA Doppler. This is consistent with previous study findings [1, 17, 18].

In the current study newborns with abnormal UA, Doppler studies were 2.3 times more likely to develop RDS and require resuscitations respectively compared to those with normal UA Doppler studies. This is comparable with other previous studies [11–13]. Neonates from abnormal UA Doppler studies group were two times more likely to require NICU admission compared to those with normal UA Doppler studies, which is comparable to other study findings [9, 12, 14].

In the current study newborns with abnormal UA Doppler studies were 2,2.5 and 2 times more likely to have low 5[th] minute APGAR score, neonatal sepsis, and neonatal hyperbilirubinemia respectively compared to those with normal UA Doppler studies. This finding is consistent with other similar studies [9, 11–13].

Concerning neonatal mortality in this study, a total of 15(8.8%) neonates died, 24.3% from those with abnormal UA Doppler studies and 4.5% from those with normal UA Doppler studies i.e. neonates from abnormal Doppler study were 4 times more likely to end up in END compared to neonates with normal UA Doppler studies. This finding is slightly higher compared to other related studies [9, 13, 14]. This might be because of the difference in the level of neonatal care in different countries and institutions as care for preterm babies is poor in low and middle-income countries [19–21]. There is only one stillbirth from the normal UA Doppler group but there is no stillbirth from abnormal UA group. This difference is not significant and is intrapartum death as patients were recruited in labour which might be related to intrapartum care and the different threshold health professional use for operative intervention for fetuses with normal and abnormal Doppler in labour. Mode of delivery and the presence of hypertension was not associated with perinatal outcomes. However, those neonates born at gestational age less than 34 weeks were more likely to require NICU admission, develop respiratory distress syndrome, and end in early neonatal death. This is perinatal morbidity and mortality associated with preterm delivery and is consistent with other study findings [9, 11–13].

The current study has its limitations. The study was limited to short term intrapartum events and neonatal outcomes during the first 7 days of neonatal life. Additionally, in the current study decision was made based on UA Doppler flow abnormality only. If possible, it would have been good to include umbilical vein and ductus venosus Doppler flow abnormality, special evaluating their impact on reducing iatrogenic preterm delivery. The study is unmatched and there is also a possibility of gestational age confounding the outcome as the study shows growth-restricted fetuses with abnormal umbilical artery born at gestational age less than 34 weeks were more likely to require NICU admission, develop respiratory distress syndrome and end in perinatal death. It would have been better if the study was done by incorporating antenatal fetal surveillance, change patterns of umbilical artery Doppler flow, and outcomes of neonates in the first month of neonatal life.

## Conclusions

The abnormal umbilical artery Doppler waveform is associated with cesarean section delivery, neonatal intensive care unit admission, respiratory distress syndrome, neonatal sepsis, neonatal hyperbilirubinemia, and early neonatal death compared to normal umbilical artery Doppler flow.

## Supporting information

**S1 Checklist. Describes a completed strobe checklist for an observational study.**
(DOCX)

## Acknowledgments

We thank midwives and physicians who helped us with patient recruitment and data collection. We are grateful to our patients for their willingness to participate in the study.

## Author Contributions

**Conceptualization:** Roba Ararso.

**Data curation:** Lemi Belay Tolu, Roba Ararso.

**Formal analysis:** Lemi Belay Tolu, Roba Ararso.

**Methodology:** Abdulfetah Abdulkadir, Yoseph Worku.

**Project administration:** Lemi Belay Tolu, Roba Ararso.

**Resources:** Roba Ararso, Abdulfetah Abdulkadir, Yoseph Worku.

**Software:** Lemi Belay Tolu, Roba Ararso.

**Supervision:** Lemi Belay Tolu, Roba Ararso, Abdulfetah Abdulkadir, Yoseph Worku.

**Validation:** Lemi Belay Tolu, Roba Ararso, Abdulfetah Abdulkadir, Garumma Tolu Feyissa, Yoseph Worku.

**Visualization:** Lemi Belay Tolu, Roba Ararso, Abdulfetah Abdulkadir, Garumma Tolu Feyissa, Yoseph Worku.

**Writing – original draft:** Roba Ararso.

**Writing – review & editing:** Lemi Belay Tolu, Garumma Tolu Feyissa.

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
