## [Decision Letter · Decision Letter 0]

5 May 2020

PONE-D-20-03283

Perinatal Outcome of Growth Restricted Fetuses with Abnormal Umbilical Artery Doppler waveforms compared to Growth Restricted Fetuses with Normal Umbilical Artery Doppler Waveforms at a tertiary referral hospital in urban Ethiopia.

PLOS ONE

Dear Dr. Belay Tolu,

Thank you for submitting your manuscript to PLOS ONE. After careful consideration, we feel that it has merit but does not fully meet PLOS ONE’s publication criteria as it currently stands. Therefore, we invite you to submit a revised version of the manuscript that addresses the points raised during the review process.

None of the reviewers reported any conflict of interest. As the reviewers have commented, the current manuscript has several methodological errors that leads the manuscript not suitable for publication in its current form. However, if the authors provides the corrections of such errors, then the manuscript will be able to be published.

We would appreciate receiving your revised manuscript by Jun 19 2020 11:59PM. To enhance the reproducibility of your results, we recommend that if applicable you deposit your laboratory protocols in protocols.io, where a protocol can be assigned its own identifier (DOI) such that it can be cited independently in the future. For instructions see: http://journals.plos.org/plosone/s/submission-guidelines#loc-laboratory-protocols

We look forward to receiving your revised manuscript.

Kind regards,

Rogelio Cruz-Martinez, Ph.D.

Academic Editor

PLOS ONE

Journal Requirements:

1. Thank you for including your funding statement; "no"

2. Please amend either the abstract on the online submission form (via Edit Submission) or the abstract in the manuscript so that they are identical.

Reviewers' comments:

Reviewer's Responses to Questions

**Comments to the Author**

1. Is the manuscript technically sound, and do the data support the conclusions?

Reviewer #1: Partly

Reviewer #2: Partly

Reviewer #3: No

2. Has the statistical analysis been performed appropriately and rigorously? 

Reviewer #1: No

Reviewer #2: Yes

Reviewer #3: No

3. Have the authors made all data underlying the findings in their manuscript fully available?

Reviewer #1: No

Reviewer #2: Yes

Reviewer #3: No

4. Is the manuscript presented in an intelligible fashion and written in standard English?

Reviewer #1: Yes

Reviewer #2: Yes

Reviewer #3: Yes

5. Review Comments to the Author

Reviewer #1: With interest, I’ve read the manuscript “Perinatal outcome of growth restricted fetuses with abnormal umbilical artery Doppler waveforms compared to growth restricted fetuses with normal umbilical artery Doppler waveforms at a tertiary hospital in urban Ethiopia”.

This is a prospective cohort study which defines the population, exposure, comparison and outcomes as follow:

Population: Pregnant women with a suspected small fetus by estimated fetal weight below the 10th centile.

Exposure: Abnormal umbilical artery Doppler

Controls: Normal umbilical artery Doppler

Outcome: several adverse perinatal outcomes

The objective of the study is to assess the perinatal outcome among suspected small fetuses exposed and non-exposed to abnormal UA Doppler.

The narrative is a little difficult to follow and the reporting somehow confusing. The overall English quality is good.

I will be dividing my revision in three main parts: reporting, critical appraisal, and general comments.

Reporting: I will be comparing the manuscript using the STROBE criteria blinded to your own list. This is to ensure I’m not bias to your results since we may have different interpretations for each given point.

Title and abstract:

1) The authors adequately mention the study design. Also, the abstract is adequately structured

Introduction:

1) Citations should be place at the end of each line that requires such, not at the end of each paragraph.

2) Introduction must be cut down in half and just stating the essential (disease, rationale, hypothesis, objective)

Methods:

1) Please state the study design at the first line of the methods section, then followed by the setting where the study was conducted, then the eligibility criteria.

2) The studies population is supposed to be pregnant women with diagnosis of suspected small-for-gestational-age fetus by estimated fetal weight measurement below the 10th percentile. This population must be adequately defined.

3) Exposures are adequate (normal and abnormal UA).

4) After eligibility criteria, variables must be stated in a narrative way, starting with outcomes, then exposures, predictors, potential confounders, effect modifiers. All in a narrative way, without headers.

5) Please describe the procedure of how pregnant women with abnormal Doppler are handle. What classification do you use? How do you end a pregnancy with abnormal UA Doppler at a certain age?

6) Describe any efforts to address potential sources of bias.

7) Sample size must be stated before statistical analysis. Which by the way, I want to congratulate the authors for adequately calculating the sample size.

8) Ethical considerations should be reduced to 2 lines maximum.

9) Statistical methods are ok. The only limitation is How did you adjust for confounders? When describing pregnancy cohorts is almost impossible not to adjust for gestational age at birth as the most important effect modifier.

Results:

1) For prospective cohorts, please state the number of individuals potentially eligible, examined for eligibility, confirmed eligible, included in the study, completing follow-up, and analyzed. Preferably, use a flow diagram.

2) Tables should be placed at the end of the manuscript and not embedded in the middle of it.

3) Continuous variables such as maternal age, gestational age... should be stated as mean and standard deviation or median and interquartile range. Continuous variables should not be only categorized or we miss the idea of the description of the population.

4) Indicate the number of participants with missing data for each variable of interest (if any)

5) State and summarize the follow-up and time of follow. A good way to do so is by describing the mean (and SD) age at diagnosis and the mean age at delivery for each group.

6) Thank you for using RRs instead of ORs. As part of the quality assessment, please, describe the adjusted RRs. You can use the Mantel-Haenzel method. The second option is to produce a multiple regression analysis adding the confounders the logistic regression.

7) Consider translating relative risk into absolute risk for the given time period of the study. This will allow readers to give a meaning idea of the magnitude of the effect.

8) Table 4. It seems clear to me that gestational age is the most important component for adverse outcome. All analyses need to be adjusted for gestational age at birth. Also, those RR with “0” are incorrect. Maybe the analysis was not robust enough. I suggest using bootstrapping to estimate those confidence intervals due to the extremely small sample.

Discussion:

1) Please, discuss more in depth the limitations of the study. Take into account the management of women with abnormal UA Doppler.

2) Conclusions must be cut by half at least.

Critical appraisal

Selection:

1) Representativeness of the exposed cohort is adequate (Abnormal UA Doppler by different definitions)

2) Selection of the non-exposed cohort is also adequate since it was drawn from the same population

3) Ascertainment of exposure is adequate because it was obtained from a structured interview.

Comparability:

1) Adequate study controls for the most important factor (Abnormal UA and normal UA). No additional factors were used

Outcome:

1) There is unclear risk of bias on assessment of outcome. There is no explanation in whether those performing the ultrasound were the ones assessing the outcomes.

2) Adequate follow-up for outcomes to occur

3) Adequate follow-up of the whole cohort. For most described adverse perinatal outcomes, a 7-day follow-up seems adequate except for neonatal death. But this does not seem to compromise the overall quality of the study.

General comments:

1) It is important to follow the same order as the STROBE tool in a narrative way.

2) Please, add an explanation of possible confounders and how were they handled.

3) Adequately define and explain the selected population (suspected SGA).

4) Use the term suspected small-for-gestational-age for all fetuses below the 10th percentile, and fetal growth restriction for those suspected SGA + abnormal Doppler.

5) Clearly state the hospital’s protocol on how FGR fetuses are handle

6) Please use mean and SD (or median and IQR) for continuous variables

7) Add the gestational age at diagnosis and the gestational age at delivery

Minor:

1) Tables should be at the end of the article

2) Use the same font type and number for the whole manuscript. Do not use bigger fonts for the headers

Reviewer #2: The manuscript by Lemi B Tolu. et al. addressed the issue on the association between Umbilical Artery Doppler and perinatal outcome in fetal growth restriction (IUGR). Placental dysfunction is one of the main complications of pregnancies and and the risk of death or neurodevelopmental impairment is high in these cases .

During the study period the authors collected 170 patients with fetal growth restriction and they were divided into fetuses with of normal umbilical artery Doppler 133 and fetuses with abnormal umbilical artery Doppler 37 and they found, On the 7th neonatal day, 129 (97%) of normal and 29 (78.4%) of abnormal umbilical artery Doppler were alive whereas 4 (3%) of normal and 9 (24.3%) of abnormal umbilical artery Doppler studies ended in early neonatal death , however although for almost 20 years, the umbilical artery (UA) has been widely accepted as the standard to identify IUGR, this assumption was based on false premises, because it extended observations that are valid in the most severe subset of IUGR fetuses to the whole group of IUGR. While UA identifies severe placental disease, if fails to pick up instances of mild placental disease, which constitute a proportion of early-onset cases, and virtually all instances of late-onset IUGR.

Evidence during the last two decades has demonstrated that SGA, as defined by a normal UA PI, contains a large proportion of fetuses with worse perinatal outcomes than normally grown fetuses

Thus, UA Doppler cannot be used as standalone criterion to differentiate IUGR from SGA.

The authors do not provide a complete outline of the current literature in this field and the articles used as references are not complete.

Major limitations are the following:

1-Introduction

It is very long and must be trimmed

2-Materials and methods:

In general, there is a important information that should be given by the authors. The specific points are:

The definition for the late and early IUGR fetuses, SGA and non SGA are required .

2.1Among the exclusion criteria did they also consider the presence of diabetes ,hypertension?

2.2 More information are required regarding the management of time of delivery, according to the local management guidelines .

2.3Was fetal well-being tests performed in all cases as standard protocol? please describe the monitoring and management protocol for IUGR

2.4Doppler results of the middle cerebral artery and uterine arteries should be added.

3-Discussion:

It should be clarified why the Doppler values of MCA and uterine artery were not taken into account for the classification and management of IUGR cases .

The following references should be added and commented in the discussion

1LINDQVIST PG, MOLIN J. Does antenatal identification of small-for-gestational age fetuses significantly improve their outcome? Ultrasound Obstet Gynecol 2005;25:258-64.

2. GARDOSI J, MADURASINGHE V, WILLIAMS M, MALIK A, FRANCIS A. Maternal and fetal risk factors for stillbirth: population based study. BMJ 2013;346:f108.

3. SKOVRON ML, BERKOWITZ GS, LAPINSKI RH, KIM JM, CHITKARA U. Evaluation of early thirdtrimester ultrasound screening for intrauterine growth retardation. J Ultrasound Med 1991;10:153-9.

4. FIGUERAS F, EIXARCH E, GRATACOS E, GARDOSI J. Predictiveness of antenatal umbilical artery Doppler for adverse pregnancy outcome in small-for-gestational-age babies according to customised birthweight centiles: population-based study. Bjog 2008;115:590-4.

5. RICHARDUS JH, GRAAFMANS WC, VERLOOVE-VANHORICK SP, MACKENBACH JP, EURONATAL INTERNATIONAL AUDIT P, EURONATAL WORKING G. Differences in perinatal mortality and suboptimal care between 10 European regions: results of an international audit. BJOG 2003;110:97-105.

6. ALFIREVIC Z, STAMPALIJA T, GYTE GM. Fetal and umbilical Doppler ultrasound in high-risk pregnancies. Cochrane Database Syst Rev 2010:CD007529.

7. SOOTHILL PW, BOBROW CS, HOLMES R. Small for gestational age is not a diagnosis. Ultrasound Obstet Gynecol 1999;13:225-8.

8. OROS D, FIGUERAS F, CRUZ-MARTINEZ R, MELER E, MUNMANY M, GRATACOS E. Longitudinal changes in uterine, umbilical and fetal cerebral Doppler indices in late-onset small-for-gestational age fetuses. Ultrasound Obstet Gynecol 2011;37:191-5.

9. DOCTOR BA, O'RIORDAN MA, KIRCHNER HL, SHAH D, HACK M. Perinatal correlates and neonatal outcomes of small for gestational age infants born at term gestation. Am J Obstet Gynecol 2001;185:652-9.

10. MCCOWAN LM, HARDING JE, STEWART AW. Umbilical artery Doppler studies in small for gestational age babies reflect disease severity. Bjog 2000;107:916-25.

11. SEVERI FM, BOCCHI C, VISENTIN A, et al. Uterine and fetal cerebral Doppler predict the outcome of third-trimester small-for-gestational age fetuses with normal umbilical artery Doppler. Ultrasound Obstet Gynecol 2002;19:225-8.

12. BAHADO-SINGH RO, KOVANCI E, JEFFRES A, et al. The Doppler cerebroplacental ratio and perinatal outcome in intrauterine growth restriction. Am J Obstet Gynecol 1999;180:7506.

Reviewer #3: PONE-D-20-03283

Perinatal Outcome of Growth Restricted Fetuses with Abnormal Umbilical Artery Doppler waveforms compared to Growth Restricted Fetuses with Normal Umbilical Artery Doppler Waveforms at a tertiary referral hospital in urban Ethiopia.

The following study describes the use of umbilical artery (UA) Doppler to predict abnormal findings and worse perinatal outcomes in intrauterine growth restriction (IUGR). The study has a good number of patients, however it lacks information which may be novel and contribute to the subject; in reality, the idea that UA Doppler is associated with worse outcomes is something that is known and this study only confirms current previously published knowledge. Furthermore, the study does not include other variables which are relevant to the study and prognosis of IUGR. I believe that the study may be relevant locally, but provides little information otherwise.

• The Introduction provides outdated information and lacks references; i.e. the concepts that UA is the only vessel used for IUGR, or the classification for symmetric and asymmetric IUGR.

• I am not sure the sample size calculation is adequate, since they claim it is 150, but usually this would be per study group. Since the proportion of IUGR with abnormal UA Doppler is rare, this would seem logical.

• The definition of IUGR is never provided, the authors claim that the diagnosis was by attendings or ObGyn residents. This would be unacceptable due to the variability and lack of expertise, and this population would have to be confirmed or supervised.

• The Methods section lacks the information for statistical analysis; Chi squared is only used for comparison of parametric proportions, however the authors present comparisons of medians(range), do not specify if they tested normality and do not describe their analysis adequately.

I would suggest the authors redefine their population because their results are very strange; for starters they have a high proportion of abnormal UA Doppler near term than preterm, which is not what commonly happens. Secondly, they do not provide basic information for delivery (Gestational age at delivery, birth weight, birth weight centile). Thirdly, they do not provide their criteria for diagnosis of the problem (IUGR) but rather assume it was previously diagnosed. I think the data requires revision and it would be more suitable for publication in a local journal.

6. PLOS authors have the option to publish the peer review history of their article (what does this mean?). If published, this will include your full peer review and any attached files.

Reviewer #1: No

Reviewer #2: No

Reviewer #3: No

---

## [Author Response · Author response to Decision Letter 0]

11 May 2020

May 10, 2020

To: PLOS ONE Editor in chief.

Dear Editor in chief.

We would like to thank the reviewers for their thoughtful review of the manuscript. They raise important issues and their inputs are very helpful for improving the manuscript. We agree with almost all their comments and we have revised our manuscript accordingly. We respond below in detail to each of the reviewer’s comments. We hope that you find our responses satisfactory and that the manuscript is now acceptable for publication 

Looking forward hearing from you soon

Sincerely,

Lemi B Tolu (MD, Assistant prof of obstetrics and gynecology).

Saint Paul’s Millennium Medical College(SPHMMC)

Department of Obstetrics and Gynecology

Addis Ababa, Ethiopia.

Email: lemi.belay@gmail.com

Dear editor and reviewer 

Thanks for thoughtful review of the manuscript. Below is point by point response to raised concerns and how we changed the manuscript according to the comments.

Editor comments:

1. Thank you for including your funding statement; "no"

Authors: Dear editor thank you very much, we have corrected as “The authors received no specific funding for this work.”

2. Please amend either the abstract on the online submission form (via Edit Submission) or the abstract in the manuscript so that they are identical.

Authors: Dear editor thank you very much, we have corrected.

Reviewer #1

Title and abstract:

1) The authors adequately mention the study design. Also, the abstract is adequately structured

Introduction:

1) Citations should be place at the end of each line that requires such, not at the end of each paragraph.

2) Introduction must be cut down in half and just stating the essential (disease, rationale, hypothesis, objective)

Authors: Dear author thank you very much, almost we rewrite introduction (page 4, lines 69-117)

Methods:

1. Please state the study design at the first line of the methods section, then followed by the setting where the study was conducted, then the eligibility criteria.

Authors: Dear reviewer corrected as per your recommendation (Page 6-7, lines 128-169).

2. The studies population is supposed to be pregnant women with diagnosis of suspected small-for-gestational-age fetus by estimated fetal weight measurement below the 10th percentile. This population must be adequately defined.

Authors: Edited and population well defined, hospital protocol included (Page 6, lines 135-140)

3. Exposures are adequate (normal and abnormal UA).

Authors: thank you very much for the comment.

4. After eligibility criteria, variables must be stated in a narrative way, starting with outcomes, then exposures, predictors, potential confounders, effect modifiers. All in a narrative way, without headers.

Authors: Dear reviewer comment well taken, and correction made (Page 8, lines 163-169)

5. Please describe the procedure of how pregnant women with abnormal Doppler are handle. What classification do you use? How do you end a pregnancy with abnormal UA Doppler at a certain age?

Authors: Dear reviewer thank you very much comment well taken, and protocol of the hospital stated on page 7, lines 135-148.

6. Describe any efforts to address potential sources of bias.

Authors: we collected data using structured questionnaire’s, data collectors were not involved in patient care and well trained on the tool, supervisors cross check data collection (Line 232-237). 

7. Sample size must be stated before statistical analysis. Which by the way, I want to congratulate the authors for adequately calculating the sample size?

Authors: Dear reviewer thank you very much, the manuscript is structed accordingly.

8. Ethical considerations should be reduced to 2 lines maximum.

Authors: Comment well taken and addressed (Page 13, 262-264) 

9. Statistical methods are ok. The only limitation is How did you adjust for confounders? When describing pregnancy cohorts is almost impossible not to adjust for gestational age at birth as the most important effect modifier.

Authors: Dear reviewer thank you very much, we also used multiple regression to control confounders.

Results:

1. For prospective cohorts, please state the number of individuals potentially eligible, examined for eligibility, confirmed eligible, included in the study, completing follow-up, and analyzed. Preferably, use a flow diagram.

Authors: Dear reviewer the comment is well taken and addressed (page 13, line 271-276) 

2. Tables should be placed at the end of the manuscript and not embedded in the middle of it.

Authors: Thank you very much and we moved tables to end of the manuscript.

3. Continuous variables such as maternal age, gestational age... should be stated as mean and standard deviation or median and interquartile range. Continuous variables should not be only categorized, or we miss the idea of the description of the population.

Authors: Dear reviewer gestational age is described in mean and average duration of follow up.

4. Indicate the number of participants with missing data for each variable of interest (if any)

Authors: Dear reviewer we didn’t come across missed data (page 13, lines 277-281) 

5. State and summarize the follow-up and time of follow. A good way to do so is by describing the mean (and SD) age at diagnosis and the mean age at delivery for each group.

Authors: Thanks, comment is well taken and addressed (Page 13, lines 277-281)

6. Thank you for using RRs instead of ORs. As part of the quality assessment, please, describe the adjusted RRs. You can use the Mantel-Haenzel method. The second option is to produce a multiple regression analysis adding the confounders the logistic regression.

Authors: Dear reviewer, thank you very much, we also multiple logistic regression.

7. Consider translating relative risk into absolute risk for the given time of the study. This will allow readers to give a meaning idea of the magnitude of the effect.

Authors: Dear reviewer thank you very much for the suggestion, but we didn’t think that translating in to absolute risk is as such applicable in the current manuscript.

8. Table 4. It seems clear to me that gestational age is the most important component for adverse outcome. All analyses need to be adjusted for gestational age at birth. Also, those RR with “0” are incorrect. Maybe the analysis was not robust enough. I suggest using bootstrapping to estimate those confidence intervals due to the extremely small sample.

Authors: Dear reviewer this is an amazing catch, loved it. Thank you very much, we have corrected it (Table 4)

Discussion:

2. Please, discuss more in depth the limitations of the study. Consider the management of women with abnormal UA Doppler.

3. Conclusions must be cut by half at least.

Authors: Dear reviewer thank you very much manuscript is modified accordingly (Page 21, lines 388-394 and Page 22, lines 400-404)

Critical appraisal

Authors: Dear reviewer thank you very much for such input which helped us to improve the manuscript.

General comments:

1. It is important to follow the same order as the STROBE tool in a narrative way.

2. Please, add an explanation of possible confounders and how were they handled.

3. Adequately define and explain the selected population (suspected SGA).

4. Use the term suspected small-for-gestational-age for all fetuses below the 10th percentile, and fetal growth restriction for those suspected SGA + abnormal Doppler.

5. Clearly state the hospital’s protocol on how FGR fetuses are handle

6. Please use mean and SD (or median and IQR) for continuous variables

7. Add the gestational age at diagnosis and the gestational age at delivery

Authors: Dear reviewer these general comments are addressed in the above explanations.

Minor:

1. Tables should be at the end of the article

2. Use the same font type and number for the whole manuscript. Do not use bigger fonts for the headers

Authors: Dear reviewer we have moved tables to the end of manuscript and the manuscript is also prepared according to PLOS ONE requirement.

Reviewer#2

General 

1. Thus, UA Doppler cannot be used as standalone criterion to differentiate IUGR from SGA.

Author: Dear reviewer we never said UA is a standalone criterion, but we said its most commonly used. MCA, UA, UV and DV are often used together. Our only aim in the current study is to compare perinatal outcome of abnormal UA to normal UA, we have also included the protocol used in our hospital (Page 7, lines 133-147). Special in low and middle-income countries Umbilical artery is very commonly used for diagnosis and follow up growth restriction.

2. The authors do not provide a complete outline of the current literature in this field and the articles used as references are not complete.

Authors: Dear reviewer we have revised the introduction part by including updated information we came across in our search.

Introduction.

1. Introduction is very long and must be trimmed

Authors: dear reviewer thanks for the comment, we have revised and rewrote introduction accordingly.

Materials and methods:

1. The definition for the late and early IUGR fetuses, SGA and non-SGA are required.

Authors: Dear reviewer the comment is well taken and addressed on page 7, lines 135-149.

2. Among the exclusion criteria did they also consider the presence of diabetes, hypertension

Authors: Dear reviewer we excluded diabetes, but we included hypertension and did multiple logistic regression to control its confounding effect.

3. More information is required regarding the management of time of delivery, according to the local management guidelines.

Authors: Dear reviewer, the comment is well taken and addressed on page 7, lines 135-148.

4. Were fetal well-being tests performed in all cases as standard protocol? please describe the monitoring and management protocol for IUGR

Authors: Dear reviewer we have included hospital protocol which includes follow up of fetal wellbeing (Page 7, lines 135-148)

5. Doppler results of the middle cerebral artery and uterine arteries should be added.

Authors: Dear reviewer in the hospital we conducted study MCA is done for patients with suspected IUGR but normal UA doppler flow to measure cerebroplacental ratio to see brain sparing which is expected to occur before umbilical artery abnormality, but not considered as universal standard practice. Uterine artery doppler is used for prediction of preeclampsia and growth restriction (predicting uteroplacental insufficiency). The hospital doesn’t use uterine artery for diagnosis or follow up of growth restriction, we didn’t come across literature suggesting such practice otherwise. Dear reviewer that’s why we focus on comparing growth restriction with normal and abnormal umbilical artery. Thanks for understanding us.

Discussion

1. It should be clarified why the Doppler values of MCA and uterine artery were not considered for the classification and management of IUGR cases.

Authors: Dear reviewer please see the explanation under question number 5.

2. The following references should be added and commented in the discussion.

Authors: Dear reviewer thank you very much we have used the following references in our manuscript from the suggestions.

a. GARDOSI J, MADURASINGHE V, WILLIAMS M, MALIK A, FRANCIS A. Maternal and fetal risk factors for stillbirth: population-based study. BMJ 2013;346: f108.

b. FIGUERAS F, EIXARCH E, GRATACOS E, GARDOSI J. Predictiveness of antenatal umbilical artery Doppler for adverse pregnancy outcome in small-for-gestational-age babies according to customised birthweight centiles: population-based study. Bjog 2008; 115:590-4.

c. ALFIREVIC Z, STAMPALIJA T, GYTE GM. Fetal and umbilical Doppler ultrasound in high-risk pregnancies. Cochrane Database Syst Rev 2010:CD007529.

d. OROS D, FIGUERAS F, CRUZ-MARTINEZ R, MELER E, MUNMANY M, GRATACOS E. Longitudinal changes in uterine, umbilical and fetal cerebral Doppler indices in late-onset small-for-gestational age fetuses. Ultrasound Obstet Gynecol 2011; 37:191-5

e. DOCTOR BA, O'RIORDAN MA, KIRCHNER HL, SHAH D, HACK M. Perinatal correlates and neonatal outcomes of small for gestational age infants born at term gestation. Am J Obstet Gynecol 2001; 185:652-9.

Reviewer #3

1. The Introduction provides outdated information and lacks references; i.e. the concepts that UA is the only vessel used for IUGR, or the classification for symmetric and asymmetric IUGR.

Authors: Dear reviewer the introduction is rewritten incorporating comments. We didn’t state UA as the only vessel, but we said its commonly used and studied. Special in low and middle-income countries umbilical artery is very commonly used for diagnosis and follow up of fetuses with growth restriction. Thanks for understanding us.

2. I am not sure the sample size calculation is adequate, since they claim it is 150, but usually this would be per study group. Since the proportion of IUGR with abnormal UA Doppler is rare, this would seem logical.

Authors: Thank you very much, that’s what we got by calculation as it stated it’s also a rare scenario too.

3. The definition of IUGR is never provided, the authors claim that the diagnosis was by attendings or ObGyn residents. This would be unacceptable due to the variability and lack of expertise, and this population would have to be confirmed or supervised

Authors: Dear reviewer the hospital protocol is included in the manuscript which has also definition of IUGR (Page 7, lines 135- 148)

4. The Methods section lacks the information for statistical analysis; Chi squared is only used for comparison of parametric proportions, however the authors present comparisons of medians(range), do not specify if they tested normality and do not describe their analysis adequately.

 Authors: Dear reviewer we used chi square, cross tabulation and regression. 

5. I would suggest the authors redefine their population because their results are very strange; for starters they have a high proportion of abnormal UA Doppler near term than preterm, which is not what commonly happens. Secondly, they do not provide basic information for delivery (Gestational age at delivery, birth weight, birth weight centile). Thirdly, they do not provide their criteria for diagnosis of the problem (IUGR) but rather assume it was previously diagnosed. 

Authors: Dear reviewer thank you very much for the comment. We have included hospital protocol for the diagnosis and follow up of growth restriction (Page 7, lines 135-148). Regarding the gestational age of term and preterm it might because what is presented is gestational age at delivery. With gestational age at delivery almost 50% are preterm and 50% are term. There is no statistical significant difference in terms of birthweight between two groups as shown in table 2.

---

## [Editor Report · Decision Letter 1]

18 May 2020

PONE-D-20-03283R1

Perinatal Outcome of Growth Restricted Fetuses with Abnormal Umbilical Artery Doppler waveforms compared to Growth Restricted Fetuses with Normal Umbilical Artery Doppler Waveforms at a tertiary referral hospital in urban Ethiopia.

PLOS ONE

Dear Dr. Tolu,

Thank you for submitting your manuscript to PLOS ONE. After careful consideration, we feel that it has merit but does not fully meet PLOS ONE’s publication criteria as it currently stands. Therefore, we invite you to submit a revised version of the manuscript that addresses the points raised during the review process.

The manuscript has been improved accordingly but it is still not suitable for publication in this Journal. It requires further corrections. 

We would appreciate receiving your revised manuscript by Jul 02 2020 11:59PM. To enhance the reproducibility of your results, we recommend that if applicable you deposit your laboratory protocols in protocols.io, where a protocol can be assigned its own identifier (DOI) such that it can be cited independently in the future. For instructions see: http://journals.plos.org/plosone/s/submission-guidelines#loc-laboratory-protocols

We look forward to receiving your revised manuscript.

Kind regards,

Rogelio Cruz-Martinez, Ph.D.

Academic Editor

PLOS ONE

Additional Editor Comments (if provided):

Firstly, the manuscript has multiple typographical and grammatical errors and thus, a more thorough editing process is necessary. Please replace “doppler” by Doppler in all the manuscript.

Normal umbilical artery Doppler should be considered as values below the 95th centile, values below the 10th centile should not be considered abnormal. An international consensus exist for definition of abnormal UA Doppler by including only the pulsatility index and therefore, UA Doppler indices such as S/D ratio and RI should be excluded.

Perinatal death should include also those cases with intrauterine fetal demise

In the abstract section, please include the p values between the study group comparisons.

“On the 7th neonatal day, 129(97%) of normal and 29(78.4%) of abnormal umbilical artery doppler were alive whereas 4(3%) of normal and 9(24.3%) of abnormal umbilical artery Doppler studies ended in early neonatal death”. Please replace this paragraph by the proportion of perinatal death between the study groups.

The frequency of adverse perinatal outcome (perinatal death, NICU admission, neonatal morbidity) should be adjusted by gestational age at birth) and all such outcomes (and not only NICU admission and neonatal death) should be also summarized in the abstract section and in the Results section.

Please specify in the Methods section how was the UA Doppler evaluated, site and angle of insonation, number of included waveforms and ultrasound settings, etc. Were all ultrasoud Doppler measurements performed at the day of delivery? Please specify.

---

## [Author Response · Author response to Decision Letter 1]

26 May 2020

May 26, 2020

To: PLOS ONE Editor in chief.

Dear Editor in chief.

We would like to thank the editor and reviewers for their thoughtful review of the manuscript. They raise important issues and their inputs are very helpful for improving the manuscript. We agree with almost all their comments and we have revised our manuscript accordingly. We respond below in detail to each of the editor comments. We hope that you find our responses satisfactory and that the manuscript is now acceptable for publication 

Looking forward hearing from you soon

Sincerely,

Lemi B Tolu (MD, Assistant prof of obstetrics and gynecology).

Saint Paul’s Millennium Medical College(SPHMMC)

Department of Obstetrics and Gynecology

Addis Ababa, Ethiopia.

Email: lemi.belay@gmail.com

Dear editor 

Thanks for thoughtful review of the manuscript. Below is point by point response to raised concerns and how we changed the manuscript according to the comments.

1. Firstly, the manuscript has multiple typographical and grammatical errors and thus, a more thorough editing process is necessary. Please replace “doppler” by Doppler in all the manuscript.

Authors: Dear editor thank you very much, two authors independently revised the typos and grammar of the manuscript and much is changed.

2. Normal umbilical artery Doppler should be considered as values below the 95th centile, values below the 10th centile should not be considered abnormal. An international consensus exists for definition of abnormal UA Doppler by including only the pulsatility index and therefore, UA Doppler indices such as S/D ratio and RI should be excluded.

Authors: Dear editor thank you very much we have corrected definition (page 7, line 135). On which indices to use, as you said PI is most recommended. But still raised S/D or absent or reversed doppler are abnormal that’s why we included in the operational definition.

3. Perinatal death should include also those cases with intrauterine fetal demise

Authors: Dear editor thank you very much, we have corrected with all appreciation, (Page 8, lines 151,152)

4. In the abstract section, please include the p values between the study group comparisons.

Authors: Dear editor thank you, corrected (abstract section)

5. “On the 7th neonatal day, 129(97%) of normal and 29(78.4%) of abnormal umbilical artery doppler were alive whereas 4(3%) of normal and 9(24.3%) of abnormal umbilical artery Doppler studies ended in early neonatal death”. Please replace this paragraph by the proportion of perinatal death between the study groups.

Authors: Dear editor thank you very much for the feedback, we have corrected (page 2, line 35, page 10 line 233) 

6. The frequency of adverse perinatal outcome (perinatal death, NICU admission, neonatal morbidity) should be adjusted by gestational age at birth) and all such outcomes (and not only NICU admission and neonatal death) should be also summarized in the abstract section and in the Results section.

Authors: Dear editor that’s very true, thanks a lot. Upon multiple regression only NICU admission, respiratory distress and perinatal death is associated with gestational age, that’s why we did for them. We have included in the abstract section as limitation of the study.

7. Please specify in the Methods section how was the UA Doppler evaluated, site and angle of insonation, number of included waveforms and ultrasound settings, etc. Were all ultrasound Doppler measurements performed at the day of delivery? Please specify.

Authors: Dear editor thank a lot. We have included what we use in the method section (Page 6 lines 102-104,114)

---

## [Editor Report · Decision Letter 2]

3 Jun 2020

Perinatal Outcome of Growth Restricted Fetuses with Abnormal Umbilical Artery Doppler waveforms compared to Growth Restricted Fetuses with Normal Umbilical Artery Doppler Waveforms at a tertiary referral hospital in urban Ethiopia.

PONE-D-20-03283R2

Dear Dr. Tolu,

We’re pleased to inform you that your manuscript has been judged scientifically suitable for publication and will be formally accepted for publication once it meets all outstanding technical requirements.

Kind regards,

Rogelio Cruz-Martinez, Ph.D.

Academic Editor

PLOS ONE

---

## [Editor Report · Acceptance letter]

9 Jun 2020

PONE-D-20-03283R2 

Perinatal Outcome of Growth Restricted Fetuses with Abnormal Umbilical Artery Doppler waveforms compared to Growth Restricted Fetuses with Normal Umbilical Artery Doppler Waveforms at a tertiary referral hospital in urban Ethiopia. 

Dear Dr. Tolu:

I'm pleased to inform you that your manuscript has been deemed suitable for publication in PLOS ONE. Congratulations! Your manuscript is now with our production department. 

Kind regards, 

on behalf of

Dr Rogelio Cruz-Martinez 

Academic Editor

PLOS ONE